# PPD: Permutation Phase Defense Against Adversarial Examples in Deep Learning

## Abstract

Deep neural networks have demonstrated cutting edge performance on various tasks including classification. However, it is well known that adversarially designed imperceptible perturbation of the input can mislead advanced classifiers. In this paper, *Permutation Phase Defense* (PPD), is proposed as a novel method to resist adversarial attacks. PPD combines random permutation of the image with phase component of its Fourier transform. The basic idea behind this approach is to turn adversarial defense problems analogously into symmetric cryptography, which relies solely on safekeeping of the keys for security. In PPD, safe keeping of the selected permutation ensures effectiveness against adversarial attacks. Testing PPD on MNIST and CIFAR-10 datasets yielded state-of-the-art robustness against the most powerful adversarial attacks currently available.

## 1 Introduction

Recent advancements in deep learning have brought Deep Neural Networks (DNNs) to security-sensitive applications such as self-driving cars, malware detection, face recognition, etc. Although DNNs provide tremendous performance in these applications, they are vulnerable to adversarial attacks (Szegedy et al., 2013; Goodfellow et al., 2014; Athalye & Sutskever, 2017). Especially in computer vision applications, small adversarial perturbation of the input can mislead the state-of-the-art DNNs while being imperceptible to human eye (Goodfellow et al., 2014; Nguyen et al., 2015; Moosavi-Dezfooli et al., 2017; Akhtar & Mian, 2018).

Since the existence of adversarial examples was discovered (Szegedy et al., 2013), numerous attacks and defenses have been proposed. Attacks typically try to push the image towards the decision boundaries of the classifier (Goodfellow et al., 2014; Carlini & Wagner, 2016; Kurakin et al., 2016; Dong et al., 2018). Defenses, on the other hand, can be categorized into three groups: (a) adversarial training, (b) hiding the classifier, and (c) hiding the input image.

**Adversarial Training.** As a natural defense, Szegedy et al. (2013); Goodfellow et al. (2014) suggested to augment the training set with adversarial examples to reshape the decision boundaries of the classifier. Adversarial training requires an attack algorithm to generate adversarial examples during training. The resulting classifier is shown to be robust against the same attack used during training (Goodfellow et al., 2014; Madry et al., 2017; Tramèr et al., 2017). However, it is still vulnerable to other attacks or even the same attack with more perturbation budget (Sharma & Chen, 2018; Madry et al., 2017).

**Hiding the Classifier.** The idea of hiding the classifier has been investigated in various ways. One naive way is to not reveal the weights of the neural network. However, adversarial examples generated by a substitute model can still fool the hidden model (Papernot et al., 2017; Liu et al., 2016), a phenomenon known as transferability of adversarial examples across different structures. Another way is to conceal the gradient of loss function to guard against attacks that take advantage of gradient (Buckman et al., 2018; Song et al., 2017; Samangouei et al., 2018; Guo et al., 2017). However, all of these defenses are circumvented by approximately recovering the gradient (Athalye et al., 2018).

**Hiding the Input Image.** Another general approach to diminish adversarial examples is to hide the input image. Following this idea, Guo et al. (2017) suggested using randomization to hide the input image by two methods: (a) image quilting: reconstruct image by replacing small patches with

clean patches from a database and (b) randomly drop pixels and recover them by total variance minimization. One fundamental restriction of these randomized defenses is that a large portion of pixels has to be preserved for correct classification. This leaves the door open for adversarial attacks. Indeed, both of these defenses are completely broken (Athalye et al., 2018).

Given that the defenses proposed under the last two categories are bypassed, to the best of our knowledge, adversarial training with examples generated by Projected Gradient Descent (PGD) attack (Madry et al., 2017) remains current state-of-the-art. However, its robustness is limited to specific attack types with certain strength. Changing the attack type or exceeding the perturbation limit degrades the accuracy significantly.

In this paper, *Permutation Phase Defense* (PPD) is proposed to mitigate adversarial examples. Following the philosophy of hiding the input image, we suggest to *"encrypt"* the image by applying two transformations before feeding the image to the neural network: First, randomly shuffle the pixels of the image with a fixed random seed hidden from the adversary. Second, convert the permuted image to the phase of its two dimensional Fourier transform (2d-DFT) (see Figure 1). This pipeline is used in both training and inference phases. The Fourier transformation captures the freqnency at which adjacent pixels of the input change along both axes. This frequency depends on the relational positions of the pixels rather than their absolute value. On the other hand, the relational positions are already hidden from the adversary by using the permutation block. The random seed plays the role of the key in cryptography.

PPD offers some advantages as follows:

- PPD outperforms current state-of-the-art defense in two ways: First, PPD is a general defense and is not restricted to specific attack types. Second, accuracy of PPD does not drop drastically by increasing perturbation budget of adversary (Table 1).

- In contrast to image quilting and random pixel drop (Guo et al. (2017)), PPD randomly shuffles all the pixels in the pixel domain. Therefore, important pixels of the phase domain remain hidden because of their dependence on the relational positions of the pixels in the pixel domain.

- Increasing adversarial perturbation budget is expected to decrease accuracy. PPD, however, can distribute perturbation over different pixels in the phase domain such that adversarial perturbation is not more effective than random noise (Figures 4 and 5).

## 2 PRELIMINARIES

### 2.1 NOTATION

Let $(\mathcal{X}, d)$ be a metric space where $\mathcal{X} = [0,1]^{H \times W \times C}$ is the image space in the pixel domain for images with height $H$, width $W$ and $C$ channels and $d : \mathcal{X} \times \mathcal{X} \to [0, \infty)$ is a metric. A classifier $h(\cdot)$ is a function from $\mathcal{X}$ to label probabilities such that $h_i(x)$ is the probability that image $x$ corresponds to label $i$. Let $c^*(x)$ be the true label of image $x$ and $c(x) = \arg\max_i h_i(x)$ be the label predicted by the classifier.

### 2.2 ADVERSARIAL EXAMPLES

Intuitively speaking, adversarial examples are distorted images that visually resemble the clean images but can fool the classifier. Thus, given a classifier $h(\cdot)$, $\epsilon > 0$ and image $x \in \mathcal{X}$, adversarial image $x' \in \mathcal{X}$ satisfies two properties: $d(x, x') \leq \epsilon$ and $c(x') \neq c^*(x)$. Ideally, metric $d$ should represent visual distance. However, there is no clear mathematical metric for visual distance. Therefore, to compare against a common ground, researchers have considered typical mathematical metrics induced by $\ell_2$ and $\ell_\infty$ norms.

Figure 1: Permutation Phase Defense (PPD): image in the pixel domain is first passed through a random permutation block with a fixed seed. The seed is fixed for all images and hidden from the adversary. The permuted image is then converted to the phase of its Fourier transform and fed to the neural network. The random permutation block conceals what pixels are neighbors, and the pixel2phase block determines values of the pixels in the phase domain based on frequency of change in neighboring pixels. This pipeline is used in both training and inference phases.

## 3 PERMUTATION PHASE DEFENSE (PPD)

### 3.1 DEFENSE STRUCTURE

We address the problem of defending against adversarial examples. We hide the input space by two transformations preceding the main neural network (see Figure 1):

- **Random permutation block:** pixels of the input image are permuted according to a fixed random permutation. The permutation seed is fixed and hidden from the adversary.
- **Pixel to phase block:** permuted image is converted to the phase of its 2-dimensional discrete Fourier transform (2d-DFT).

These two blocks precede the neural network in both training and testing phases.

Mathematically speaking, consider channel $c$ of image $x \in \mathcal{X}$. Its 2d-DFT can be written as

$$F_{lk} = \frac{1}{\sqrt{HW}} \sum_{h=0}^{H-1} \sum_{w=0}^{W-1} x_{hw} e^{-j2\pi(\frac{l}{H}h + \frac{k}{W}w)} \tag{1}$$

where $l = 0, \cdots, H-1$ and $k = 0, \cdots, W-1$ and $x_{hw}$ is $(h, w)$ image pixel value. $F_{lk}$ is a complex number that can be represented in the polar coordinates as $F_{lk} = A_{lk} \exp(j\varphi_{lk})$ where $A_{lk} \in [0, \infty)$ is the magnitude and $\varphi_{lk} \in [-\pi, \pi)$ is the phase of $F_{lk}$. Let $\mathcal{P} = [-\pi, \pi)^{H \times W \times C}$ be the phase space and $p : \mathcal{X} \to \mathcal{P}$ be a function that maps each channel of the image in the pixel domain to its phase. Let $\sigma : \mathcal{X} \to \mathcal{X}$ be a channel-wise permutation, i.e., pixels in each channel of the input image are permuted according to a fixed permutation. Note that same permutation is used for all channels and all images. The PPD classifier can then be written as

$$h(x) = f(p(\sigma(x))) \tag{2}$$

where $f(\cdot)$ is the neural network.

### 3.2 INTUITION BEHIND THE DEFENSE

It is known that random noise is unable to fool a trained neural network (Fawzi et al., 2016). Motivated by this fact, if we train the neural network in a domain that is hidden from adversary, adversarial perturbations will not affect more than random noise in the actual domain fed to the neural network and thus not fool the classifier.

Following this idea, Fourier transformation is used to capture the frequency at which neighboring pixels change. Indeed, the values of the image in the Fourier domain do not purely depend on the values of pixels in the pixel domain. Instead, they depend on what pixels are adjacent in the pixel domain, while neighboring pixels are hidden from adversary by the random permutation block. Therefore, adversary will have little hope to attack the classifier due to almost no knowledge about the input space.

#### 3.2.1 WHY RANDOM PERMUTATION IS NOT ENOUGH?

We would like to emphasize that input transformations that partially hide the input space are not secure. For example, *total variance minimization* (randomly drop pixels and replace them with total

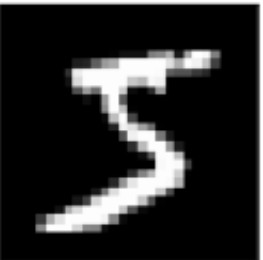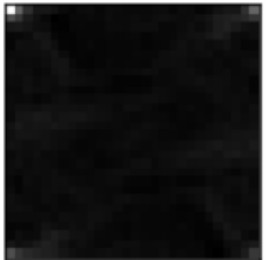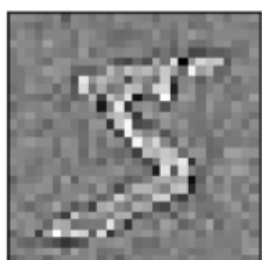

Figure 2: Phase component of 2d-DFT contains key information of the image. Thus, neural networks can be trained in the phase domain rather than the original pixel domain. (Left) original image. (Middle) image reconstructed from magnitude only by setting phase to zero. It almost has no information of the original image. (Right) image reconstructed from phase only by setting magnitude to unity. Edges are preserved and main features of the original image are restored.

variance minimization), *image quilting* (replacing patches of the input image by patches from clean images) (Guo et al., 2017) are all defeated (Athalye et al., 2018). One possible explanation for this failure is that although the adversary does not have full knowledge of the actual image fed to the neural network (by using randomness in dropping pixels and patching), good portion of the image is still preserved. Hence, it is possible for adversary to attack.

Similarly, we have observed that random permutation by itself (without the pixel2phase block) is not sufficient to defeat adversary. Adversarial images generated by attacking a vanilla neural network can substantially decrease the accuracy of a *permuted* network. This is because important pixels (e.g. white pixels in MNIST) in the original domain will still remain important pixels in the *permuted* domain. Therefore, regardless of the employed permutation, if adversary attacks important pixels of the original domain (e.g. by converting white pixels to gray pixels in MNIST), it has successfully attacked important pixels in the permuted domain.

### 3.2.2 WHY PHASE OF 2D-DFT IS USED?

It is well known that phase component of Fourier transform is crucial in visual perception of human (Oppenheim & Lim, 1980; Morrone & Burr, 1988). Indeed, an image reconstructed from phase only preserves essential information about the edges (see Figure 2). This suggests that phase carries key information about the image and thus it is possible to train neural networks to operate in the phase domain rather than the pixel domain. One might ask what if both phase and magnitude are fed to the neural network by doubling the size of the input space? Our effort to train such a model was not fruitful but we leave it as a future research direction.

## 4 EXPERIMENTS

In this section, we test PPD trained on MNIST and CIFAR-10 against state-of-the-art adversarial attacks. MNIST is a dataset of 28x28 grayscale images of handwritten digits and consists of 60,000 training as well as 10,000 test images. CIFAR-10 dataset includes 50,000 32x32 color training images in 10 classes as well as 10,000 test images. We assume that full information about the defense architecture is available to the adversary except for the permutation seed. In addition, adversary can probe the classifier with its crafted images and use output probabilities to construct more complicated adversarial examples.

Unknown permutation seed prevents adversary from taking advantage of classifier parameters, because if a wrong permutation seed is used, the classifier acts like a random-guess classifier. So far we have not come up with an effective way to test PPD against attacks that have access to model parameters. Therefore, we leave it as an untested claim that *PPD is secure against attacks with full knowledge of classifier parameters as long as the permutation seed is not revealed.*

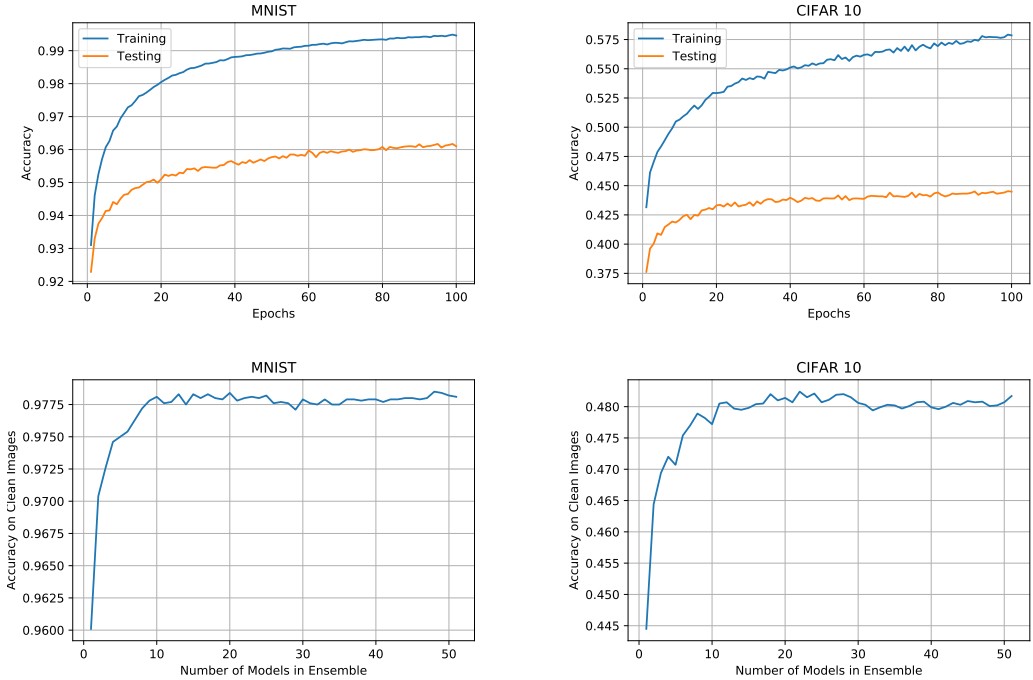

Figure 3: (Top) training and testing accuracy of a PPD model on clean images. Pixels of each image are permuted according to a fixed permutation, then phase component of the 2d-DFT of the permuted image is fed to the neural network. A 3-layer dense neural network can achieve 96% out of sample accuracy on MNIST and 45% on CIFAR 10. (Bottom) accuracy of ensemble of models on clean test images. Each model in the ensemble is trained for a different permutation. Ensemble of 10 PPD models can reach above 97.75% accuracy on clean MNIST test images and around 48% accuracy on clean CIFAR-10 test images.

Furthermore, huge space of permutations precludes the possibility of guessing the permutation seed. For 28x28 MNIST images, there are $784! > 10^{1500}$ possible permutations which is significantly more than the number of atoms in the universe [1].

## 4.1 TRAINING PPD CLASSIFIER ON MNIST AND CIFAR-10

PPD requires training the neural network in the *permutation-phase* domain. Before evaluating the defense against attacks, we need to ensure that high accuracy on clean images is indeed achievable. This is a challenging task. The analysis in Figure 2 suggests that training in the phase domain must be feasible in principle since the crucial information of the image are preserved. However, a permutation block preceding the pixel2phase block may make training difficult. In this section, we successfully train a 3-layer dense neural network of size 800x300x10 on MNIST and CIFAR-10 datasets to attain 96% and 45% test accuracy, respectively (see Figure 3). Moreover, our experiments show that ensemble of 10 models (with different permutation seeds) yields around 98% and 48% (see Figure 3) accuracy on clean test images of MNIST and CIFAR-10, respectively. Note that the goal here is not to achieve the best possible classifier trained in the permutation-phase domain, rather to show that even with a naive 3-layer dense neural network without any tricks such as data augmentation, deep residual nets, etc., it is possible to reach certain level of learning. Moreover, as shown in the following subsections, even with such simple networks, PPD can attain the state-of-the-art performance on adversarial examples on both MNIST and CIFAR-10 datasets. We believe that better results on clean images automatically translate to better results on adversarial examples and encourage research community to start a new line of research on how to train more accurate classifiers in the permutation-phase domain.

---

[1]Currently, it is estimated that there are around $10^{80}$ atoms in the universe.

## 4.2 ROBUSTNESS OF PPD AGAINST ATTACKS

Now let's see how PPD behaves against state-of-the-art attacks. Cleverhans library v2.1.0 (Papernot et al., 2018) is used for attack implementations. Each attack requires a set of parameters as input to control the algorithm as well as the perturbation strength. Various set of parameters for different attacks make it difficult to compare attacks against each other. To overcome this issue, we decided to define an average perturbation measure to evaluate adversary's strength and provide a unified measure to compare attacks against each other:

$$\frac{1}{n} \sum_{i=1}^{n} \|x_i - x_i'\| \tag{3}$$

where $x_i$ is the original image (vectorized) and $x_i'$ is the adversarial image (vectorized). The norm used in this measure can be any norm, but we focus on $\ell_\infty$ and $\ell_2$ norms (and call corresponding perturbations $\ell_\infty$ and $\ell_2$-perturbation, respectively) here as most attacks are specifically designed to perform well in these norms. However, robustness of PPD is not restricted to specific type of norms. Average perturbation measure is essential to make claims about a defense. To elaborate more on this point, suppose that adversary can change any pixel by 0.5, which means it can modify any image to a fully gray image and fool any classifier completely. Therefore, to quantify the robustness of defenses, it is crucial to mention how much perturbation adversary is allowed to make. Guo et al. (2017) considered $\frac{1}{n} \sum_{i=1}^{n} \frac{\|x_i - x_i'\|_2}{\|x_i\|_2}$ as a perturbation measure. However, division by $\|x_i\|_2$ makes the perturbation measure asymmetric for dark (low value of $\|x_i\|_2$) and light (high value of $\|x_i\|_2$) images.

*Implementation Detail:* A 3-layer dense neural network of size 800x300x10 is used to train on MNIST and CIFAR-10 datasets. The first two layers are followed by relu activation and the output layer is followed by softmax. Pixels are normalized to $[0, 1]$ for both datasets. The pixel2phase block is implemented in Tensorflow so that the derivative of this transformation is available for attacks to use. Given a certain level of perturbation strength, attacks parameters were tuned to provide maximum rate of fooling on the model used by adversary to generate adversarial examples. For example, for iterative attacks such as MIM, BIM, PGD and CW, number of iterations was increased to a level above which it could not fool the network any further. Parameter tuning provides a fair comparison among different attacks of certain strength and is missing from previous work. Thus, in addition to reporting attack parameters, we encourage researchers to evaluate attacks using perturbation measure in (3) and then fine tune parameters to get the best fooling rate under the fixed distortion level. This gives a common ground to compare different attacks.

*Adversary's Knowledge:* The permutation seed is not revealed to the adversary. Thus, adversary is granted knowledge of training dataset and parameters (such as learning rate, optimizer, regularization parameters, etc.). Moreover, it can probe the network with input images of it's choice and receive the output label probabilities.

### 4.2.1 ATTACKS

Adversaries try to limit perturbation with respect to specific norms with the goal of keeping the adversarial modification imperceptible to human eye. Most adversarial attacks proposed so far are designed to keep $\ell_\infty$ or $\ell_2$ perturbations limited. Thus, we restrict our evaluations to these two groups by considering FGSM (Goodfellow et al., 2014), BIM (Kurakin et al., 2016), MIM (Dong et al., 2018) and $\ell_\infty$-PGD (Madry et al., 2017) as $\ell_\infty$ oriented attacks and CW (Carlini & Wagner, 2016), $\ell_2$-PGD (Madry et al., 2017), $\ell_2$-FGM (Goodfellow et al., 2014) as $\ell_2$ oriented attacks. However, we should emphasize that PPD robustness is not restricted to distortions with specific norms. Conceptually, it is supposed to provide more general robustness. This is in particular important because previously proposed adversarial training using $\ell_\infty$-PGD (Madry et al., 2017) showed robustness to $\ell_\infty$ attacks of the same strength, but was broken by $\ell_1$ oriented attacks (Sharma & Chen, 2018). Other attacks such as LBFGS (Szegedy et al., 2013), DeepFool (Moosavi-Dezfooli et al., 2016), JSMA (Papernot et al., 2016) and EoT (Athalye & Sutskever, 2017) were not effective on PPD, so we ignored reporting the results here. In addition, our efforts to attack PPD in the phase domain (i.e., adversary generates adversarial phase using a neural network trained in the phase of a permuted image and then combine the adversarial phase with benign magnitude to recover adversarial image in the pixel domain) were not fruitful, so we are not reporting them either.

Table 1: Accuracy of ensemble of 50 PPD models on $\ell_\infty$ and $\ell_2$ attacks for MNIST and CIFAR-10. The lowest accuracy for each perturbation strength is in bold.

| | MNIST | | | | | CIFAR-10 | | | | |
|---|---|---|---|---|---|---|---|---|---|---|
| $\ell_\infty$ pert. | 0.03 | 0.1 | 0.2 | 0.3 | 0.4 | 0.03 | 0.1 | 0.2 | 0.3 | 0.4 |
| FGSM | 97.8% | 97.6% | 97.1% | 95.4% | 91% | 48.2% | 47.9% | 45.3% | 39.7% | 31.1 |
| BIM | 97.8% | 97.6% | 96.7% | 95.2% | 91% | 48.2% | 47.7% | 45.2% | 39% | 30.1% |
| PGD | 97.8% | 97.7% | 97% | 95.2% | 91.3% | 48.3% | 47.6% | 45.4% | 41.6% | 37.1% |
| MIM | 97.8% | 97.6% | **95.8%** | **87.4%** | **67.5%** | 48.2% | 47.5% | 40.4% | **26.1%** | **15.6%** |
| Blackbox | **97.6 %** | **97.4 %** | 95.9% | 92.1% | 84.1% | **47.6%** | **45.1%** | **36.7%** | 26.4% | 18.5% |
| $\ell_2$ pert. | 0.1 | 0.7 | 1.1 | 3.2 | 4 | 0.3 | 2 | 4 | 6.5 | 10.5 |
| FGM | 97.8% | 97.8% | 97.8% | 97.3% | 97.1% | 48.1% | 48.1% | 48% | 47.3% | 45.4% |
| PGD | 97.8% | 97.8% | 97.8% | **97.3%** | **96.7%** | 48.3% | 48.3% | **47.5%** | **45.7%** | **39.3%** |
| CW | **97.7%** | **97.4%** | **97.7%** | 97.7% | 97.8% | **48%** | **48%** | 47.9% | 46.5% | 39.5% |

To get a sense of perturbation strength, note that if adversary can modify each pixel by 0.5, it can fool any classifier by converting the image to a complete gray image. This translates to $\ell_\infty$ distortion of 0.5, and $\ell_2$ distortion of 14 and 27.71 for MNIST and CIFAR-10, respectively [2]. Moreover, it is worth to compare the results with images distorted by random noise of same strength. Random noise cannot fool the neural networks (Fawzi et al., 2016). Thus, it gives a good understanding of how effective attacks are. $\ell_\infty$ random noise of strength $p$ is generated by adding $\pm p$ uniformly at random to each pixel. To generate $\ell_2$ random noise of strength $p$, a matrix with the same size as the image is generated with elements chosen iid from $[-0.5, 0.5)$. This matrix is then scaled (to satisfy $\ell_2$ norm of $p$) and added to the image.

Figures 4 and 5 show performance of PPD against $\ell_\infty$ and $\ell_2$ attacks, respectively. A group of 51 PPD models (each for a different permutation seed) is trained on training data. Adversary is given one of these PPD models as well as the test data to generate adversarial examples. The resulting adversarial images are then fed to the ensemble of other models to evaluate accuracy. This setup is to meet the assumption that permutation seed is hidden from adversary. Figures 4 and 5 show that except for large perturbations, adversarial attacks have not been more destructive than random noise distortion. This supports the claim that PPD essentially hides salient pixels from adversary in a way that adversarial perturbation simply acts like random noise. Note that although we perform experiments on 50 PPD models, ensemble of 10 of them is sufficient to achieve this level of accuracy (see Figure 3). Another attack scenario is Blackbox (Papernot et al., 2017) where adversary probes an ensemble of PPD models as a black box and tries to train a substitute model to mimic decision boundaries of the target model. The substitute model is then used to craft adversarial images. We found that Blackbox attack is effective on MNIST dataset. Thus, a standard denoising preprocessing is used before feeding to PPD. Table 1 provides corresponding numerical values.

### 4.2.2 Comparison with Madry et al. (2017)

As a solid step to understand and guard against adversarial examples, Madry et al. (2017) proposed PGD as possibly the strongest $\ell_\infty$ attack and claimed that adversarial training with examples generated by PGD secures the classifier against all $\ell_\infty$ attacks of the same strength. Given that some other defenses (Dhillon et al., 2018; Kannan et al., 2018) are circumvented (Athalye et al., 2018; Engstrom et al., 2018), to the best of our knowledge, adversarial training with PGD gives the current state-of-the-art on both MNIST and CIFAR-10, achieving 88.8% accuracy on $\ell_\infty$ attack of strength 0.3 for MNIST [3] and 44.7% on $\ell_\infty$ attack of strength 0.03 on CIFAR-10 [4]. With the same attack budget, our worst accuracy on MNIST and CIFAR-10 are 87.4% and 47.6%, respectively (Table 1).

Increasing $\ell_\infty$ attack budget to 0.4 for MNIST and 0.1 for CIFAR-10 brings Madry's accuracy down to 0% and 10%, respectively (Figure 6 of Madry et al. (2017)). This is while PPD still gives worst case accuracy of 67.5% and 45.1%, respectively.

---

[2]Calculation detail: (1) MNIST: $\sqrt{28 \times 28 \times (0.5)^2} = 14$ and (2) CIFAR-10: $\sqrt{32 \times 32 \times 3 \times (0.5)^2} = 27.71$.

[3]MNIST challenge

[4]CIFAR-10 challenge

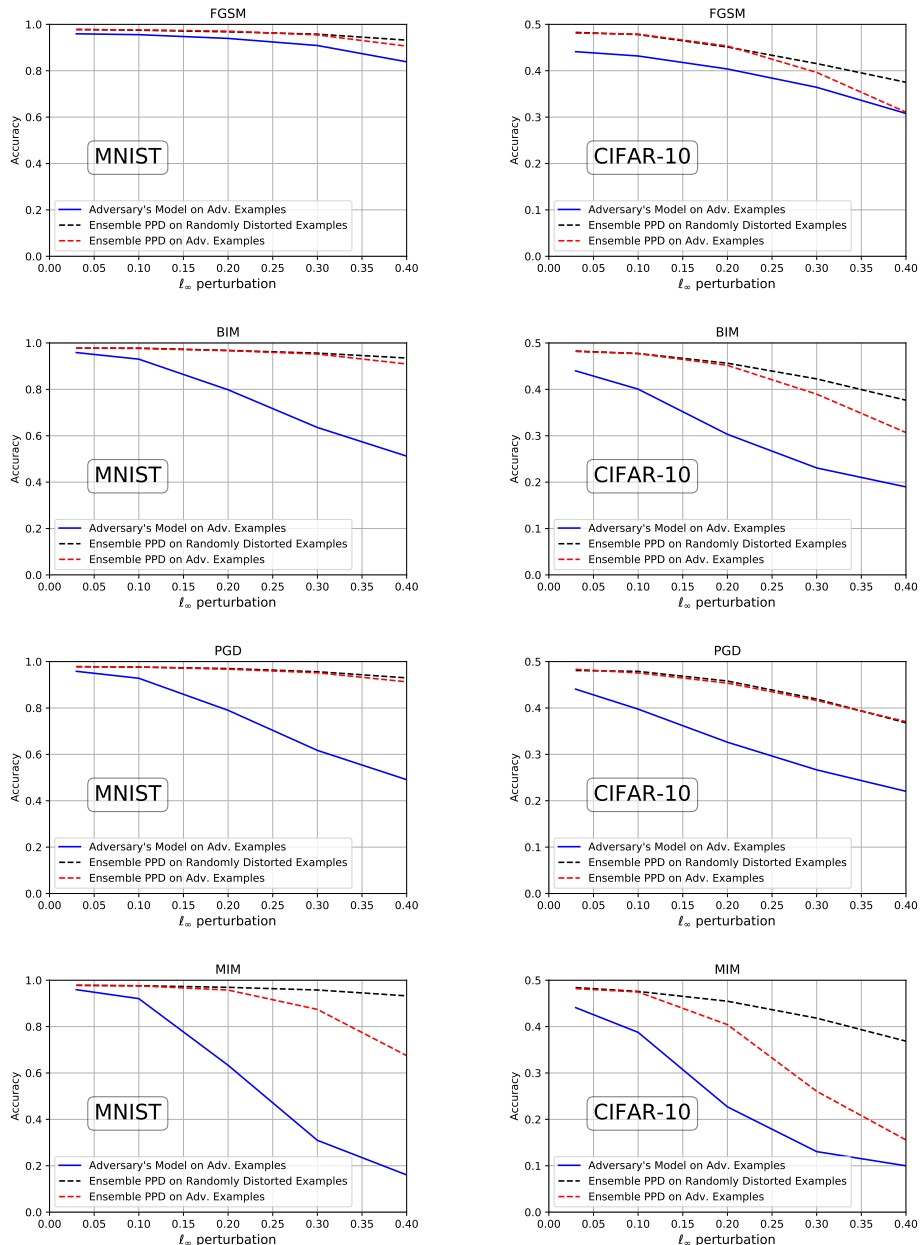

Figure 4: Performance of ensemble of 50 PPD models against different $\ell_\infty$ attacks. Left column shows results on MNIST, right column on CIFAR-10. Adversary uses a PPD model with wrong permutation seed to generate adversarial examples (solid blue line). Adversarial examples are then fed to the ensemble of other 50 models (dashed red line). Performance of ensemble of 50 PPD models on images distorted by uniform noise is shown for a benchmark comparison (dashed black line). On MNIST, it can be seen that except for MIM with large $\ell_\infty$ perturbation, adversarial perturbations cannot fool PPD more than random noise. On CIFAR-10, for $\ell_\infty$ perturbations less than 0.1, adversarial attacks are not more effective than random noise distortion. See Figure 6 in the Appendix to confirm that perturbations larger than 0.1 on CIFAR-10 can make it difficult even for human eye to classify correctly.

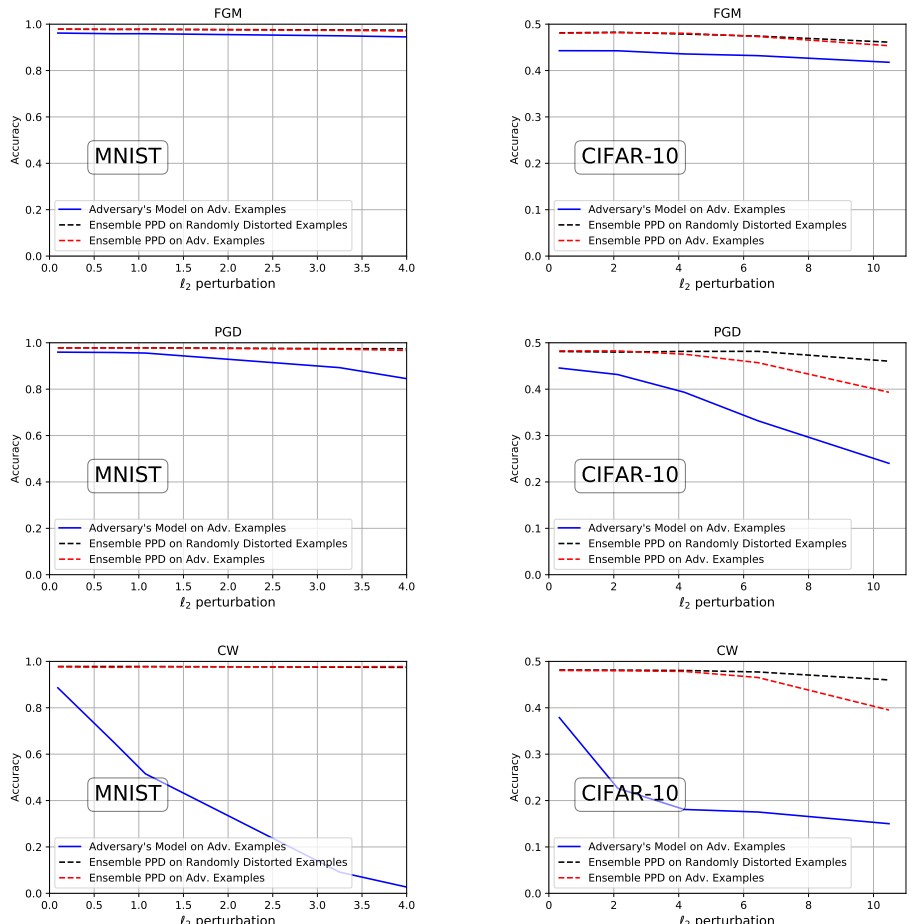

Figure 5: Performance of ensemble of 50 PPD models against different $\ell_2$ attacks. Left column shows results on MNIST, right column on CIFAR-10. Adversary uses a PPD model with wrong permutation seed to generate adversarial examples (solid blue line). Adversarial examples are then fed to the ensemble of other 50 models (dashed red line). Performance of ensemble of 50 PPD models on images distorted by uniform noise is shown for a benchmark comparison (dashed black line). It can be seen that for $\ell_2$ perturbations less than 4, adversarial attacks on PPD are not more effective than random noise distortion. See Figure 6 in the Appendix to visualize perturbed images.

On the other hand, Madry's defense is not robust against $\ell_1$ (Sharma & Chen, 2017) and $\ell_2$ attacks. In particular, $\ell_2$ attack of strength 0.35 can decrease Madry's accuracy down to 10% on CIFAR-10 (Figure 6 of Madry et al. (2017)). This is while PPD retains significant resistance to $\ell_2$ perturbations (Table 1).

## 5 CONCLUSION

In this paper, we proposed a novel approach that combines random permutations and piexel2phase as a solid defense against a large collection of recent powerful adversarial attacks. Inspired by symmetric cryptography, which relies merely on safekeeping of the keys for security, concealing the selected permutation ensures the effectiveness of our adversarial defense. We demonstrated the effectiveness of our approach by performing extensive experiments on the MNIST dataset and tried to extend this approach to the CIFAR-10 dataset with competitive results. We argue that our approach can easily create multiple models that correspond to different permutations. In addition, using ensemble of multiple models for final prediction is an inevitable cost for an effective defense.

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

APPENDIX

A. VISUALIZATION OF PERTURBED IMAGES

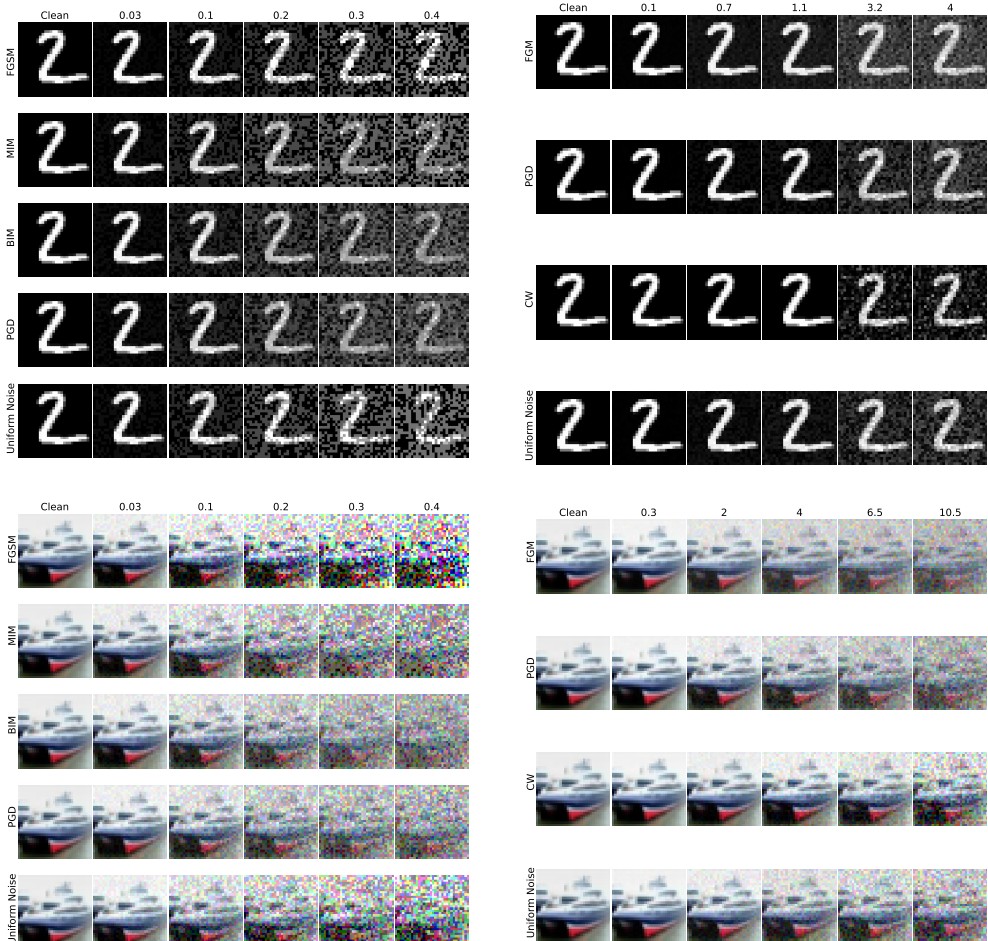

Figure 6: MNIST image of digit 2 and CIFAR-10 image of a ship distorted by adversarial attacks and random noise. Left group of columns is for $\ell_\infty$ perturbation, right group denotes $\ell_2$ perturbation. Each row corresponds to a different attack. As moving from left to the right, adversarial perturbation is increased. Last row shows results for random noise distortion with corresponding strength. Note that $\ell_\infty$ perturbation larger than 0.1 and $\ell_2$ perturbation larger than 6.5 make it difficult even for human eye to correctly classify CIFAR-10 images.

