# OpenReview forum: "PPD: Permutation Phase Defense Against Adversarial Examples in Deep Learning"
_ICLR.cc/2019/Conference_

### Official Review · AnonReviewer1 · 2018-11-01
**Good idea but has limited use case**

**Rating:** 4
**Confidence:** 5

**Review:**

This paper explores the idea of utilizing a secret random permutation in the Fourier phase domain to defense against adversarial examples. The idea is drawn from cryptography, where the random permutation is treated as a secret key that the adversarial does not have access to. This setting has practical limitations, but is plausible in theory.

While the defense technique is certainly novel and inspired, its use case seems limited to simple datasets such as MNIST. The permuted phase component does not admit weight sharing and invariances exploited by convolutional networks, which results in severely hindered clean accuracy -- only 96% on MNIST and 45% on CIFAR-10 for a single model. While the security of a model against adversarial attacks is important, a defense should not sacrifice clean accuracy to such an extent. For this weakness, I recommend rejection but encourage the authors to continue exploring in this direction for a more suitable scheme that does not compromise clean accuracy.

Pros:
- Novel defense technique against very challenging white-box attacks.
- Sound threat model drawn from traditional security.
- Clearly written.

Cons:
- Poor clean accuracy makes the technique very impractical.
- Insufficient baselines. While the permutation is kept as a secret, it is plausible that the adversary may attempt to learn the transformation when given enough input-output pairs. Also, the adversary may attack an ensemble of PPD models for different random permutations (i.e. expectation over random permutations). The authors should introduce an appropriate threat model and evaluate this defense against plausible attacks under that threat model.

---

> ### Author Response · Authors · 2018-11-26
> **Reply: Good idea but has limited use case**
>
> We thank the reviewer for the detailed feedback and address the comments below:
>
> Reviewer comment: Poor clean accuracy makes the technique very impractical.
>
> Our response: The 48% accuracy on CIFAR-10 is for a simple 3 layer dense neural network and our goal was to show that even with such a simple network, SOTA robustness can be achieved. We believe that high accuracy combined with adversarial robustness is possible for CIFAR-10, and transfer learning shows promise in this direction. What we plan to do as future work is to replace the neural network in the PPD pipeline with a pre-trained model on massive datasets such as ImageNet and retrain the final layers to fit the permutation-phase domain.
>
> Reviewer comment: Insufficient baselines. While the permutation is kept as a secret, it is plausible that the adversary may attempt to learn the transformation when given enough input-output pairs.
>
> Our response: Thanks for bringing this attack scenario to our attention. To test PPD against an adversary that tries to learn the transformation, we used Blackbox attack (https://arxiv.org/abs/1602.02697 ). In this attack, adversary probes an ensemble of PPD models as a black box by enough input-output pairs and trains a substitute model. The substitute model is then used to craft adversarial examples. Table 1 is updated with the Blackbox results.
>
> Reviewer comment: The adversary may attack an ensemble of PPD models for different random permutations (i.e., expectation over random permutations). The authors should introduce an appropriate threat model and evaluate this defense against plausible attacks under that threat model.
>
> Our response: Per the reviewer's request, we tested PPD against expectation over transformation (EoT) (https://arxiv.org/abs/1707.07397) where the permutation is considered as the transformation. 30 PPD models (with different permutations) are used for EoT. The adversarial examples are then fed to an ensemble of 10 PPD models (with different permutations from the 30 models). Our experiments show that EoT can not decrease accuracy more than an adversary that attacks with a single model. One possible explanation is that EoT is mostly useful in the case that sampling a few transformations provides a good approximation of the expectation over transformation. For example, two scenarios that EoT is shown to be successful are: (1) synthesizing adversarial examples that are robust to camera viewpoint shift and (2) breaking a defense that randomly drops pixels of the image and replaces them with total variance minimization. In both of these two scenarios, sampling a few transformations gives a good idea of the expectation. However, in PPD, each transformation has its own fingerprint which is totally different from others.

---

### Official Review · AnonReviewer3 · 2018-11-02
**Interesting approach.**

**Rating:** 7
**Confidence:** 4

**Review:**

This paper proposes Permutation Phase Defense (PPD), a novel image hiding method to resist adversarial attacks. PPD relies on safekeeping of the key, specifically the seed used for permuting the image pixels. The paper demonstrated the method on MNIST and CIFAR10, and evaluates it against a number of adversarial attacks. The method appears to be robust across attacks and distortion levels.

The idea is clearly presented and evaluated.

*Details to Improve*
It would be interesting to see how performance degrades if the opponent trains with an ensemble of random keys.

It would be great to see this extended to convolutional networks.

---

> ### Author Response · Authors · 2018-11-26
> **Reply: Intersting approach**
>
> We thank the reviewer for the positive feedback.
>
> Reviewer comment: It would be interesting to see how performance degrades if the opponent trains with an ensemble of random keys.
>
> Our response: Per the reviewer requested, we tested PPD against expectation over transformation (EoT) attack (https://arxiv.org/abs/1707.07397 ). EoT uses an ensemble of 30 PPD models to make adversarial examples. Our experiments showed that EoT could not degrade the performance more than an adversary that uses a single PPD model. One possible explanation is that each permutation yields a unique domain. In other words, information gained by other domains does not reveal much about the unknown domains.
>
> Reviewer comment: It would be great to see this extended to convolutional networks.
>
> Our response: We have observed that PPD shows robustness even in the case that convolutional networks are used. However, the accuracy in the convolutional networks is slightly worse than dense networks. For example, convolutional nets achieved around 97% (rather than 98%) on MNIST dataset and 44% (rather than 48%) on CIFAR-10 dataset. One possible explanation for this observation is that the permutation block breaks the local properties of the images exploited by convolutional networks.

---

### Official Review · AnonReviewer2 · 2018-11-06

**Rating:** 6
**Confidence:** 3

**Review:**

The Paper is written rather well and addresses relevant research questions.
In summary the authors propose a  simple and intuitive method to improve the defense on adversarial attacks by combining random permutations and using a 2d DFT. The experiments with regards to robustness to adversarial attacks I find convincing, however the overall performance is not very good (such as the accuracy on Cifar10).

My main points of critique are:

1. The test accuracy on Cifar10 seems to be quite low,  due to the permutation of the inputs. This
makes me question  how favorable the trade-off between robustness vs performance is.

2. The authors state "We believe that better results on clean images automatically translate to better results on adversarial examples"

I am not sure if this is true.   One counter argument is  that better results on clean images can be obtained by memorizing more structure of the data (see [1]). But if more memorizing (as opposed to generalization) happens, the classifier is more easily fooled (the decision boundary is more complicated and exploitable).



[1] Zhang, C., Bengio, S., Hardt, M., Recht, B., & Vinyals, O. (2016). Understanding deep learning requires rethinking generalization. arXiv preprint arXiv:1611.03530.

---

> ### Author Response · Authors · 2018-11-26
> **Reply: No Title**
>
> We thank the reviewer for the valuable feedback and address the comments in the following:
>
> Reviewer comment: The test accuracy on Cifar10 seems to be quite low,  due to the permutation of the inputs. This makes me question  how favorable the trade-off between robustness vs performance is.
>
> Our response: Training on CIFAR-10 is a much more complicated task compared to MNIST and moving to the permutation-phase domain makes it even more difficult. We don't know at the moment what type of network structure and learning technique results in the best accuracy in the permutation-phase domain, but our experiments demonstrate that even a simple 3 layer dense neural network that achieves 48\% accuracy on clean test images, provides SOTA robustness. We believe that using techniques such as transfer learning helps to improve accuracy on CIFAR-10 data set, but this idea requires more time and resources to evaluate and we leave it for future work.
>
> Reviewer comment: The authors state "We believe that better results on clean images automatically translate to better results on adversarial examples". I am not sure if this is true. One counter argument is  that better results on clean images can be obtained by memorizing more structure of the data (see [1]). But if more memorizing (as opposed to generalization) happens, the classifier is more easily fooled (the decision boundary is more complicated and exploitable).
>
> Our response: By better results on clean images, we mean better results on clean "test" images. The paper referred by the reviewer states the ability of neural networks to memorize the entire training data set and reaching 100% training accuracy while achieving only random guess on testing data set. This is absolutely correct and in fact we have seen it even in the permutation-phase domain that training accuracy can reach as high as 100% while generalizing poorly on testing data set (although not reported in the paper).  However, our claim states that a PPD model that can reach high clean "test" accuracy, will not perform poorly on adversarial examples. This is simply because PPD breaks adversarial perturbation of pixel domain to random noise in the permutation-phase domain. Thus, the only goal left for future work is to increase clean test accuracy.

---

### Public Comment · (anonymous) · 2018-10-02
**Typo**

Section 2 title should be "Preliminaries"

---

> ### Author Response · Authors · 2018-10-10
> **Will be fixed in the revision**
>
> Thanks for pointing that out. It will be fixed in the revision.

---

### Public Comment · (anonymous) · 2018-10-02
**Abstract should be revised**

The abstract says that the defense provides "state-of-the-art robustness against the most powerful adversarial attacks currently available." This should be revised to say "the most powerful black box adversarial attacks" because all of the evaluations are against black box attacks. These are not the most powerful; white box attacks are more powerful.

---

> ### Author Response · Authors · 2018-10-10
> **Term "black box" usually describes models that hide everything**
>
> The word "black box" is not used because PPD only requires to hide the random permutation seed. Everything else can be revealed. However, the term "black box" is usually meant for models that hide structure, parameters, training dataset, etc., and can only be queried.

---

> > ### Public Comment · (anonymous) · 2018-10-10
> > **At best it can be called "grey box"?**
> >
> > If the permutation is known, the paper itself shows that the accuracy can be brought down. How then does this work as a defense against the "most powerful attacks" currently available?
> >
> > This argument of hiding the permutation is tantamount to security through obscurity.  Perhaps "most powerful grey-box attacks" is the right term, so as to not over-claim the results?
> >
> > Also, the comparison with Madry is unfair. Only meaningful attacks are those designed for the specific defense.

---

### Public Comment · (anonymous) · 2018-10-02
**Checking for gradient masking**

Many proposed defenses actually achieve only "gradient masking": they break the optimizer used for traditional attacks, but don't actually move the decision boundary.

- Have you checked that your attacks reach ~100% success rate against the target model (the one with the known permutation)? If the attacks do not mostly succeed here, this suggests that the attack fails because the new operations make optimization difficult, not because the permutation key is unknown. Sorry if this is in the paper and I've missed it.

- Have you tested your model by running ~1000 noisy perturbations for each example and picking the most damaging noisy permutation? This is essentially random search for adversarial examples and it sometimes works in cases where gradient-based search fails due to gradient masking.

---

> ### Author Response · Authors · 2018-10-10
> **PPD robustness is because of unknown permutation**
>
> -The blue curves in Figures 4 and 5 show the accuracy for the target model (the one with the known permutation). As seen in these figures, some attacks such as MIM and CW have decreased the accuracy of the target model to below 20% for MNIST. However, they have not been successful on models with hidden permutation (red dashed curves). This shows that robustness is actually achieved by unknown permutation and not gradient masking.
>
> - I'm not sure if I understood your second part of comment. Do you mean the following?
> "Adversary trains ~1000 PPD models and craft adversarial examples for each of those models. Then, tests adversarial examples of each of those models against the unknown PPD model to see which PPD model out of the 1000 models was the most damaging to the unknown PPD model?"

---

> > ### Public Comment · (anonymous) · 2018-10-10
> > **Gradient Masking vs Obscurity**
> >
> > Do you then agree that the defense purely relies on obscurity? It has been shown that some other defenses based on randomized perturbations of the input can be broken by adaptive attacks.

---

> > > ### Author Response · Authors · 2018-10-10
> > > **PPD is not just obscurity, it completely changes the input space**
> > >
> > > Randomized perturbation of the input proposed in previous work does not completely change the input space. For example, the methods proposed in this paper https://openreview.net/forum?id=SyJ7ClWCb can be broken because they just change some parts of the input through randomization. In fact, they have to keep a large portion roughly unchanged to retain accuracy. However, PPD completely changes the input space. Note that the role of pixel2phase block is to build the input image of the neural network out of the relational positions of the pixels while the relational positions are hidden using the permutation.

---

> > > > ### Public Comment · (anonymous) · 2018-10-10
> > > > **Partial vs Complete Obscurity?**
> > > >
> > > > The defense only works if you obscure how exactly you are changing the input space. What is the important of parts of the input vs the whole input?

---

> > > > > ### Author Response · Authors · 2018-10-12
> > > > > **An example for clarification**
> > > > >
> > > > > Maybe the following example helps to clarify partial vs complete obscurity:
> > > > >
> > > > > One method proposed in https://openreview.net/forum?id=SyJ7ClWCb is total variance minimization. This approach randomly selects a "small" set of pixels, and reconstructs the simplest image that is consistent with the selected pixels. Why small set of pixels? Because the neural network wants to decide based on the reconstructed image and if a lot of pixels are changed, it will fail to make a good decision. This is what I mean by saying that important parts of the image (or in other words a large portion of the image) are preserved in total variance minimization.
> > > > >
> > > > > However, PPD completely obscures the input space by random permutation + pixel2phase. Why are we saying completely hides important pixels of the image? Consider an MNIST image of digit 2. Clearly, important pixels are the white pixels in the image. If we just use random permutation (which is a partial obscurity), we are changing positions of these white pixels. So, if adversary attacks white pixels in the unpermuted image of 2 (by converting them to grey pixels), it has successfully attacked a permuted image no matter what permutation is used, because grey pixels of unpermuted image remain gray pixels in the permuted domain as well. But adding pixel2phase block after the permutation block solves this issue. Fourier transform captures frequency of change in pixel values. So, instead of looking at what pixels are white in the permuted image of digit 2, it looks at the change in pixel values of neighbors (and this is hidden from adversary due to the random permutation). So severity of attack is mitigated in this way

---

### Public Comment · (anonymous) · 2018-10-02
**Does PPD work in "gray box" settings?**

I understand that PPD is meant to work when the permutation is kept secret.
Does it still work if the attacker gets to send queries to the model and observe the output? Or can the attacker reverse-engineer the function represented by the network with few queries?
(e.g. would it stand up to this attack? https://arxiv.org/abs/1602.02697 )

---

> ### Author Response · Authors · 2018-10-10
> **Will be tested**
>
> Thanks for bringing this attack to our attention. We will test and add results to the revision.

---

### Public Comment · (anonymous) · 2018-10-10
**It seems to me that your fundamental assumption is invalid.  Or did I miss something?**

A neural network is fed with input images that could be clean or adversarial which is unknown.
Input images should go through the same pipeline.  In your case the pipeline is PPD in Figure 1.

Adversarial images generated by whatever algorithms in traditional ways (i.e., based on clean images)
or any benign test images sent to the network will go through the same permutation and Pixel2Phase in your PDD.
Therefore, attackers don't need to know your secret permutation at all.

It seems that your fundamental assumption in section 4.2 (the "Adversary’s Knowledge" paragraph)
that "yet a different permutation to craft adversarial examples" is invalid.  Or did I miss something?

---

> ### Author Response · Authors · 2018-10-10
> **Assumption is valid**
>
> Thanks for your comment. Input image by itself is not enough to craft adversarial images. Adversary requires both input image and a classifier to generate adversarial examples. However, hiding permutation seed stops adversary from using the true model (note that hiding permutation stops adversary's access to the gradient of loss function with respect to the input). So, adversary cannot push the image towards the decision boundaries of the classifier.
>
> As an alternative, adversary has to use a substitute model to craft adversarial images. Assuming that the permutation is not revealed, adversary may use a similar model trained with a different permutation which is shown to be ineffective.

---

> > ### Public Comment · (anonymous) · 2018-10-10
> > **Blackbox attacks, universal adversarial perturbations, etc**
> >
> > How does the defense fare against blackbox attacks and attacks such as universal adversarial perturbations? It seems to me that attacks that rely on the current model may fail (because of the hidden permutation) but other attacks such as the ones mentioned may succeed (accuracy similar to or less than that of Madry et al.)

---

> > > ### Author Response · Authors · 2018-10-10
> > > **Will be tested**
> > >
> > > If I understood correctly, you mean:
> > > "attacks that query the true model and construct a substitute model based on the query results without any permutation or pixel2phase block. Then, use the substitute model to craft adversarial examples."
> > > Another comment also mentioned such a scenario based on this paper:  https://arxiv.org/abs/1602.02697. We agree that this is a valid scenario for adversarial attack. It will be tested and the results will be added to the revision.

---

> > > > ### Public Comment · (anonymous) · 2018-10-10
> > > > **Simpler attacks can reveal things about your defense**
> > > >
> > > > Nope. I meant an even simpler attacks --
> > > >
> > > > Blackbox: use FGSM with a large-ish epsilon on a model without any defenses to generate adversarial examples and test your model+defense against that. This is a sure sign that the defense is based on gradient obscuring.
> > > >
> > > > Universal Adversarial Perturbations: https://arxiv.org/abs/1610.08401
> > > >
> > > > Compare both vs Madry et al. and see if you notice any improvement.

---

> > ### Public Comment · (anonymous) · 2018-10-10
> > **I feel that you did not really address my comments**
> >
> > Thanks for your quick response.  However, can you spend more time to read and think about my comments?  No matter an adversary uses or does not use a similar/same model to generate adversarial images, they don't need to perform permutation and Pixel2Phase by themselves.  Again, the raw inputs to your or any other network could be clean or adversarial images and the fact is unknown (otherwise, no problem needs to be addressed).   The inputs will go through the same pipeline and whatever secret or public permutations that you use.  The attackers don't need to know the secret permutation.  In other words, the "encryption" concept is not applicable here.  I won't further comment on your paper unless you or other readers convincingly dispute my comments or point out what I really missed in your paper.

---

> > > ### Author Response · Authors · 2018-10-12
> > > **Re: I feel that you did not really address my comments**
> > >
> > > Encryption is not happening in the inference stage. It is happening in building the adversarial images. Maybe the word "encryption" has been misleading. Sorry for that. Let me explain in other words:
> > >
> > > I understand that when you are given an image for inference you do not know whether it is adversarial or clean and you just feed it to the classifier. But the question is can adversary really generate adversarial images? That's where we are stopping adversary not in the inference stage. Adversary requires two elements to generate adversarial examples: (a) input space, and (b) classifier. But how PPD stops adversary? You can think about it in two ways:
> > >
> > > 1. If you consider the neural network as the classifier, PPD is hiding the input space.
> > > 2. If you consider the whole pipeline as the classifier, PPD is hiding the classifier.
> > >
> > > I hope you find this explanation helpful.

---

> > > > ### Public Comment · (anonymous) · 2018-10-15
> > > > **"can adversary really generate adversarial images?"**
> > > >
> > > > It has been shown repeatedly that the adversary does not need access to the gradients to construct images. A black-box attack that fools an ensemble of models would definitely be "adversarial" to your defense, I think. Worth a shot?
> > > >
> > > > https://arxiv.org/abs/1611.02770 may be worth a read if you disagree :)

---

### Public Comment · (anonymous) · 2018-10-10
**A couple of concerns**

- Your CIFAR-10 accuracy is <45%. This is half (!) of what a SOTA neural network achieves, and is *lower* than the black-box adversarial accuracy of SOTA models.

- With an L2 distortion of 4 on MNIST, it is often possible to actually change the underlying digit to a human, yet the paper claims 96%+ accuracy. What is your accuracy at a L2 distortion of larger values (6? 8?)? This is sufficiently high to modify any digit to any other digit, and should be 0%. In particular, the mean minimum distance between an MNIST digit and a different digit in a different class is about 7.5.

- Similarly, with a CIFAR-10 distortion of 0.2 (that is, 51/255), it is possible to make images completely unrecognizable to humans, yet you still claim only a two percentage point loss in accuracy. This is highly suspicious. For each test sample, it would be useful to try 10,000 different values of uniform random noise with mangitude 0.2 as suggested by Athalye et al. (2018) as a way to verify your defense is not just breaking the optimizers. You do report "On CIFAR-10, for l_\infty perturbations less than 0.1, adversarial attacks are not more effective than random noise distortion" but this does not repeatedly trying random noise for the same input sample, which is often much more effective.

- Have you tried performing an attack using Expectation over Transforms (Athalye et al. 2018) where you take the expectation over the different random seeds?

---

> ### Author Response · Authors · 2018-11-26
> **Re: A couple of concerns**
>
> Thanks for your detailed comments and sorry for the late reply. We had to perform some experiments to answer your questions.
>
> - The accuracy reported for CIFAR10 is for a simple 3 layer dense network. We agree that this is far from desirable, but we believe that if future work can reach higher accuracy on clean images in the permutation-phase domain with more advanced techniques, this automatically results in higher accuracy on adversarial examples.
>
> - The high accuracy is possibly due to the fact that the attacks do not try to modify the digit, rather try to destruct the pixels. In addition, we do not claim retaining this level of accuracy under any type of attack. The purpose of the distortion measure is to compare attacks based on the same ground. Indeed, based on the facts that you mentioned (average L2 distortion of 7.5), this shows that current attacks are not using the distortion budget efficiently.
>
> - As you mentioned, from figure 6 in the appendix, it can be seen that distortion level of 0.2 is quite large for human eye to correctly classify. The code will be released after the final decision to reproduce the results. I'm not sure if I understood your point on 10,000 different random noise. Do you mean expectation over transformation attack?
>
> - We have tested expectation over transformation (https://arxiv.org/pdf/1707.07397.pdf) by attacking a set of 30 PPD models and then test on ensemble of another 10 models. However, the results showed that no more effect than a single model.

---

### Public Comment · (anonymous) · 2018-10-10
**Can the secret permutation be extracted?**

This method develops a fixed permutation to use for all images. The results show that if an attacker learns the permutation, the method is insecure. It is therefore important that it is not possible for the attacker to learn the permutation. Have you thought at all about if this attack might be possible?

In crypto (which the paper gives as inspiration for a secret key), papers dedicate a significant amount of effort to demonstrating that the secret key is not leaked. Even under a chosen plaintext attack, it should not be possible to learn anything about the key that is being used.

---

> ### Author Response · Authors · 2018-10-12
> **Reply**
>
> Thanks for your interesting question. So far, we have not been able to come up with an attack that can learn the permutation. We know that there are huge possible permutations (784! > 10^1500) for MNIST for example. This rules out the possibility of random guess, but does not preclude the possibility of leaking information. We don't know at the moment if this kind of attack is possible.

---

### Public Comment · (anonymous) · 2018-11-14
**FFT after performing a permutation?**

It doesn't make intuitive sense why you would perform an FFT after permuting the pixels. This seems like it would destroy all spatial locality making the FFT useless.

Do you instead mean that you *first* perform a FFT and then *second* permute the pixels?

---

> ### Author Response · Authors · 2018-11-26
> **Re: FFT after performing a permutation?**
>
> Thanks for your question.
> FFT should be placed second. The role of FFT is to build an image based on the relational positions of the permuted image (while the relational positions are encrypted using the permutation block). If FFT is placed first, the whole intuition behind the defense fails because adversary can also simply first take FFT and then perform the attack. Note that random permutation by itself is not secure (see Section 3.2 and the subsections).

---

### Meta-Review · Area_Chair1 · 2018-12-15
**A novel approach but has significant issues**

**Confidence:** 5
**Recommendation:** Reject

**Metareview:**

This paper presents a new defense against adversarial examples using random permutations and a Fourier transform. The technique is clearly novel, and the paper is clearly written.

However, as the reviewers and commenters pointed out, there is a significant degradation in natural accuracy, which does not seem to be easily recoverable. This degradation is due to the random permutation of the images, which effectively disallows the use of convolutions.

Furthermore, Reviewer 1 points out that the baselines are insufficient, as the authors do not explore (a) learning the transformation, or (b) using expectation over transformation to attack the model.

This concern is further validated by the fact that Black-box attacks are often the best-performing, which is a sign of gradient masking. The authors try to address this by performing an attack against an ensemble of models, and against a substitute model attack. However, attacking an ensemble is not equivalent to optimizing the expectation, which would require sampling a new permutation at each step.

The paper thus requires significantly stronger baselines and attacks.